



**Data mining-based machine learning methods for improving**
**hydrological data: a case study of salinity field in the Western**
**Arctic Ocean**
**Shuhao Tao[1,2], Ling Du[1,2], Jiahao Li[1,2]**
*[1]Frontier Science Center for Deep Ocean Multispheres and Earth System (FDOMES)*
*and Physical Oceanography Laboratory, Ocean University of China, Qingdao, China,*
*[2]College of Oceanic and Atmospheric Sciences, Ocean University of China, Qingdao,*
*China*
**Correspondence to:** Ling Du (duling@ouc.edu.cn)



**Abstract.** In the Western Arctic Ocean lies the largest freshwater reservoir in the Arctic
Ocean, the Beaufort Gyre. Long-term changes in freshwater reservoirs are critical for
understanding the Arctic Ocean, and data from various sources, particularly measured
or reanalyzed data, must be used to the greatest extent possible. Over the past two
decades, a large number of intensive field observations and ship surveys have been
conducted in the western Arctic Ocean to obtain a large amount of CTD data. Multiple
machine learning methods were evaluated and merged to reconstruct annual salinity
product in the western Arctic Ocean over the period 2003-2022. Data mining-based
machine learning methods make use of variables determined by physical processes,
such as sea level pressure, sea ice concentration, and drift. Our objective is to effectively
manage the mean root mean square error (RMSE) of sea surface salinity, which exhibits
greater susceptibility to atmospheric, sea ice, and oceanic changes. Considering the
higher susceptibility of sea surface salinity to atmospheric, sea ice, and oceanic changes,
which leads to greater variability, we ensured that the average root mean square error
of CTD and EN4 sea surface salinity field during the machine learning training process
was constrained within 0.25psu. The machine learning process reveals that the
uncertainty in predicting sea surface salinity, as constrained by CTD data, is 0.24%,
whereas when constrained by EN4 data it reduces to 0.02%. During data merging and
post-calibrating, the weight coefficients are constrained by imposing limitations on the
uncertainty value. Compared with commonly used EN4 and ORAS5 salinity in the
Arctic Ocean, our salinity product provide more accurate descriptions of freshwater
content in the Beaufort Gyre and depth variations at its halocline base. The application
potential of this multi-machine learning results approach for evaluating and integrating
extends beyond the salinity field, encompassing hydrometeorology, sea ice thickness,
polar biogeochemistry, and other related fields. The datasets are available at
https://zenodo.org/records/10990138 (Tao and Du, 2024).

## 1. Introduction

Unlike the low- and mid-latitude oceans, the Arctic Ocean is characterized by its
extensive sea ice coverage and near-freezing sea surface water. Variations in salinity in
the Western Arctic Ocean have profound implications for stratification strength, ocean
circulation patterns, and biogeochemical cycles (Carmack et al., 2016; Cornish et al.,
2020). Freshwater reservoirs and their evolution, which are closely related to the change
of seawater salinity, have become the focus of research in the Arctic Ocean. Therefore,
obtaining accurate salinity data holds great significance for our understanding of this
unique marine environment. The mean density structure and wind-driven surface



circulation in the Arctic Ocean are predominantly influenced by two key factors: The
anti-cyclonic Beaufort Gyre located in the Canadian Basin and the Transpolar Drift
(Hall et al., 2022). Furthermore, within Western Arctic Oceans, significant amounts of
freshwater accumulate within the Beaufort Gyre. The release of this freshwater exerts
a substantial impact on local climate dynamics as well as global climate change at large
scales (Carmack et al., 2008; Giles et al., 2012; Proshutinsky et al., 2009, 2019). Our
research specifically focuses on a case study of investigating salinity product improved
by multi-machine learning results evaluating and integrating within Western Arctic
Oceans.
The presence of sea ice severely limits the availability of salinity data in the Arctic
Ocean, posing significant challenges to meeting the demands of current research.
Shipborne observations of CTD and ITP data are sporadic, posing challenges in
obtaining reliable salinity measurements. The accuracy of both model and reanalysis
data is frequently subpar. Behrentdt et al. (2018) collected a large amount of measured
data to form a Unified Database for Arctic and Subarctic Hydrography for the period
1980-2015, however, hydrological data for recent years are lacking.. In recent years,
however, highly developed measurement techniques were especially designed for
operation in the Arctic environment. Furthermore, an increasing number of research
activities and international collaboration - such as Beaufort Gyre Exploration Project
(BGEP) has generated a large number of hydrographic data in the Western Arctic ocean
and the subarctic seas (e.g., Rabe et al., 2014).
The advancement of stochastic computer science and technology in recent years has led
to an increasing utilization of machine learning methods across various domains. The
utilization of data mining-based machine learning techniques for data generation is
explored in this paper, with a focus on the salinity observed in the Western Arctic Ocean.
Machine learning techniques have already demonstrated their efficacy in data
generation tasks. For instance, Wang et al. (2023) employed a machine-learning-based
regression method to reconstruct long-term (2003-2020) sea surface pCO2 in the South
China Sea, while Chen et al. (2024) utilized the Random Forest Algorithm to generate
datasets of stable isotopes of precipitation in the Eurasian continent. The utilization of
machine learning offers distinct advantages during data reconstruction processes
including high automation, exceptional accuracy, robust scalability, and expedited
processing compared to assimilation approaches. Consequently, this paper employs
several machine learning methods to produce dependable salinity data in the western
Arctic Ocean.
We performed machine learning training on sea level pressure, sea ice concentration,





sea ice motion, as well as a large number of quality-controlled CTD data and EN4 data
using various machine learning methods. The datasets were merged to generate a
salinity product with a resolution of 0.5×0.25° above 1000m for the period spanning
from 2003 to 2022, encompassing a total of 48 vertical layers. The machine learning
performance was assessed not only through RMSE, but also by evaluating the
uncertainty resulting from data merging and post-calibrating processes. The ORAS5
and EN4 datasets were employed to investigate the Beaufort Gyre and Arctic Ocean
(Hall et al.,2022). The accuracy and reliability of our salinity product were
demonstrated by comparing it with EN4 and ORAS5 data, as well as measured
freshwater content and halocline base depth in the Beaufort Gyre region.
**2. Data and methodology**
**2.1 Study area**
The Western Arctic Ocean (140°E-120°W, 68°N-90°N) spans a vast territory with the
Beaufort Gyre, the largest fresh water reservoir in the Arctic Ocean (Fig. 1). In the
Western Arctic Ocean, sea ice covers the area in winter, while in summer, a large area
of sea ice at low latitudes melts. However, sea ice still exists in the multi-year ice zone
in the northeast of Canada Basin. The Western Arctic Ocean is mainly influenced by
the anticyclonic Beaufort High. In the western part of the Arctic Ocean, there is the
main circulation system of the Arctic Ocean, the Beaufort Gyre, which accumulates a
large amount of fresh water. The Strength of the Beaufort Gyre has been continuously
increasing, reaching a stable state after 2007, with changes in freshwater content
consistent with the strength of the gyre (Regan et al., 2019). The range of the Beaufort
Gyre expanded westward from 2003 to 2013, and contracted eastward back to the
Canadian Basin after 2014 (Lin et al., 2023). Freshwater accumulation, storage, and
release from the BG exert far-reaching impacts on both regional and global climate
systems. Therefore, accurate salinity data is very important for our study of Beaufort
Gyre.

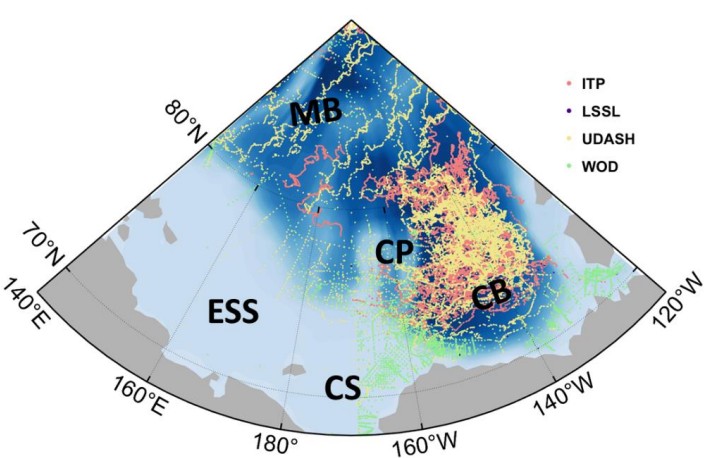


**Figure1. Topography of the Western Arctic Ocean. The map also includes the Canada Basin (CB), Chukchi sea (CS), the Chukchi Plateau (CP), East Siberian Sea (ESS) and Makarov Basin (MB).**

Our goal is to generate a set of salinity product that can be used to analyze the physical ocean environment changes in the Arctic Ocean in recent years. The procedure of improving salinity product is mainly divided into four major parts, which are data selecting, machine learning training, data merging, and post-calibrating.



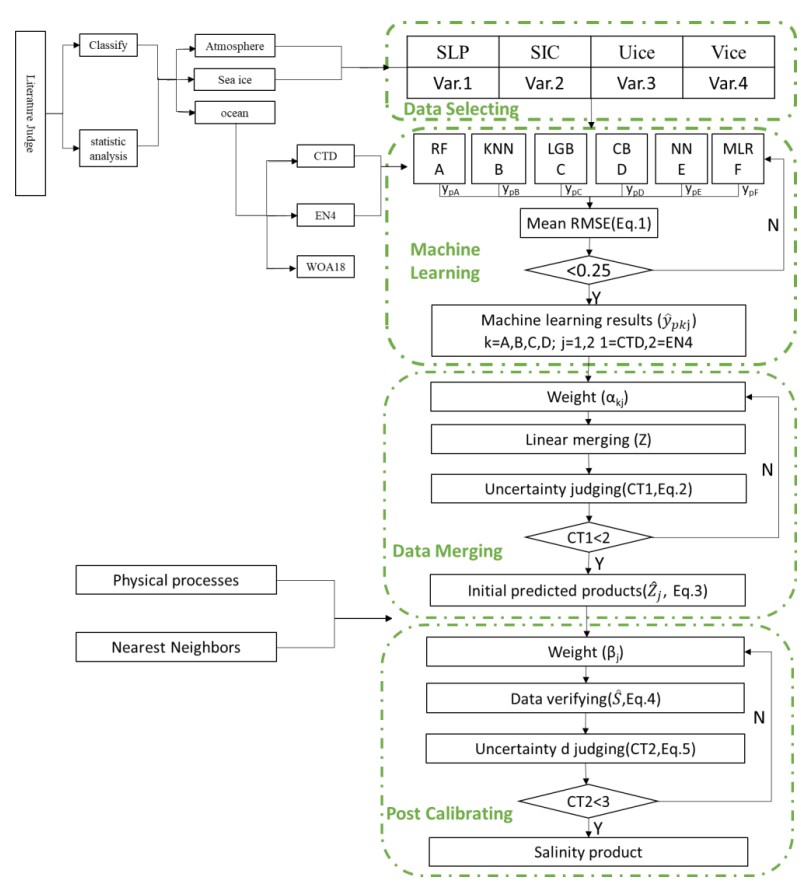


**Figure.2 Procedure for improving the salinity field in the Western Arctic Ocean**

**through a data mining-based machine learning method**



**2.2 Data Selecting**
We have collected a large amount of CTD salinity data. The World Ocean Database
(WOD) is world's largest collection of uniformly formatted, quality controlled, publicly
available ocean profile data (https://www.ncei.noaa.gov/access/world-ocean-
database/bin/getwodyearlydata.pl?Go=TimeSorted, last access: 8 December 2023). We
selected the WOD18 salinity profiles and retained the data with flags 0 and 1 based on
the quality control provided by the data itself. Unified Database for Arctic and Subarctic
Hydrography (UDASH) is a unified and high-quality temperature and salinity data set
for the Arctic Ocean and the subpolar seas north of 65° N for the period 1980-2015
(https://essd.copernicus.org/articles/10/1119/2018/, last access: 8 December 2023). Sea
ice presents a significant impediment to sustained observation of the Arctic Ocean.

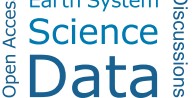



Researchers designed and field tested an automated, easily-deployed Ice-Tethered
Profiler (ITP) for Arctic study. Building on the ongoing success of ice drifters that
support multiple discrete subsurface sensors on tethers and the WHOI-developed
Moored Profiler instrument capable of moving along a tether to sample at better than
1-m vertical resolution (https://www2.whoi.edu/site/itp/data/, last access: 8 December
2023). Shipboard hydrographic data and water sampling measured on board the CCGS
Louis S. St-Laurent (LSSL) are carried out at about 30 standard sites on each cruise
(https://www2.whoi.edu/site/beaufortgyre/data/ctd-and-geochemistry/, Last access: 8
December 2023), the CTD data of LSSL collected during the 2004 expedition was not
utilized.

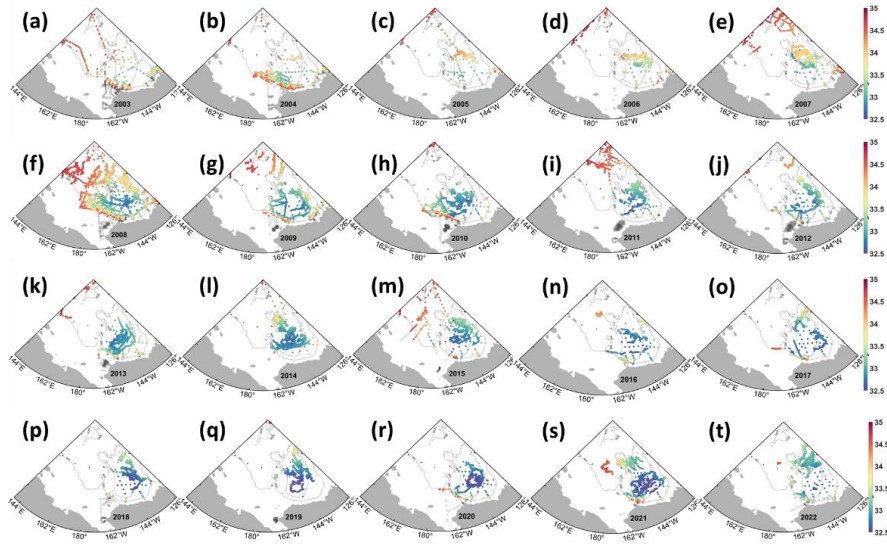


**Figure 3. Annual sea surface salinity fields from 2003 to 2022 in the Western Arctic**
**Ocean.**
The data collected include a variety of issues such as missing values, outliers, and
duplicates as well as gaps in dates and missing or incorrect latitude and longitude
information. Therefore, the collected raw data underwent pre-processing and data
cleaning. Missing data were interpolated, entries that could not be completed were
removed, and duplicate data were eliminated. This article interpolates all data onto the
WOD vertical grid in depth. The most CTD data was collected in late summer and early
autumn (August to October), while the least CTD data was collected in June. The
measured data is mainly concentrated in the Canadian Basin, with very few measured



data in the East Siberian Sea (Fig. 2). After 2003, ITP and LSSL supplemented a large
amount of CTD data in situ, so we hope to generate gridded data from 2003 to 2022.
In addition to a large amount of observed CTD data, considering the temporal and
spatial   discontinuity   of   the   observed   data,   we   have   introduced   EN4
(https://www.metoffice.gov.uk/hadobs/en4/, Last access: 8 December 2023) reanalysis
data. Furthermore, taking into account the influence of the atmosphere and sea ice on
the   ocean,   we   have   also   incorporated   SLP   data   from   ERA5
(https://cds.climate.copernicus.eu/cdsapp#!/dataset/reanalysis-era5-single-levels-
monthly-means?tab=form, Last access: 8 December 2023) and sea ice concentration
and sea ice drift field data from NSIDC (https://nsidc.org/home, Last access: 8
December 2023). We use monthly salinity data provided by the European Centre for
Medium-Range Weather Forecasts (ECMWF) through the Ocean Reanalysis System's
version 5 (ORAS5), which uses the Nucleus for European Modeling of the Ocean
(NEMOv3.4) for its ocean model coupled with a sea ice model to assess the accuracy
of salinity product. In the data selecting section, we summarized previous literature and
selected the sea level pressure field, sea ice concentration, and sea ice drift field data of
the Western Arctic Ocean as training variables for machine learning.

**2.3 Machine learning**

In the second part of the machine learning training section, we selected six commonly
used machine learning methods, which are Random Forest (RF), K Nearest Neighbor
(KNN), LightGBM (LGB), CatBoost (CB), Neural Network (NN), and Multilinear
Regression (MLR). We determined the optimal value of different machine learning
algorithm   using   optuna   hyper   parameter   methods   (code   from
https://github.com/optuna/, last access: 20 March 2024) and GridSearchCV (from
scikit-learning) for the training set. We trained EN4 and CTD data with six different
machine learning methods respectively.
It is necessary to evaluate the accuracy of any model based on certain error metrics
before applying it to specific scenarios. Common model evaluation metrics include
MAE, RMSE. The mean squared error (MSE) is the standard deviation of the residuals
(prediction error), and the residuals are the distances between the fitted line and the data
points (i.e., the residuals show the degree of concentration of the reconstructed data
around the regression line). In regression analysis, RMSE is commonly used to verify
experimental results. To assess bias, the RMSE needs to combine the magnitude of the
model data and is calculated as follows:

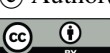

$RMSE_{Kj} = \sqrt{\frac{1}{n}\sum_{i=1}^{n}\left(y_{iKj} - y_{piKj}\right)^2}$ , $(Eq.1{:})$
where n is the number of data points; K represents different machine learning
algorithms, and there are six types in total, which are RF, KNN, LGB, CB, NN, MLR;
j=1 represents CTD data, j=2 represents EN4 data; y is the training target data; $y_p$ is
the prediction result after machine learning training.
Taking the results from 2008 of Random Forest results as an example (Fig.4), we found
that the salinity prediction at a depth of 200m is better than the prediction at the surface
(15m), and the prediction using EN4 data is better than using CTD data. However, what
is exciting is that even for the weakest prediction ability of CTD at the surface, the
RMSE is less than 0.35psu. Therefore, our evaluation of the model learning results will
mainly focus on the surface with larger prediction errors by RMSE.

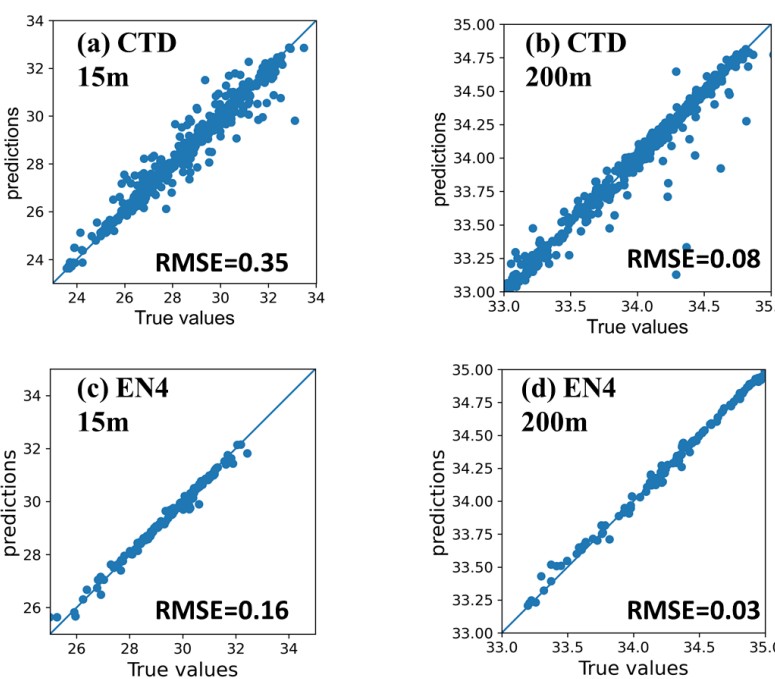


**Figure 4. Comparisons between the predicted salinity and train target salinity**
**values for the Random Forest testing pool in 2008.**
In addition to RF, we also evaluated the prediction results of surface salinity for five





other machine learning methods using RMSE (Table1), which is calculated as follows:
**Table1. Evaluation of predicted surface salinity using different machine learning**
**methods**

| | Random Forest | | K Nearest Neigbor | | LightGBM | | Catboost | | Multilinear Regression | | Neural Network | |
|---|---|---|---|---|---|---|---|---|---|---|---|---|
| | CTD | EN4 | CTD | EN4 | CTD | EN4 | CTD | EN4 | CTD | EN4 | CTD | EN4 |
| 2003 | 0.45 | 0.07 | 0.49 | 0 | 0.43 | 0.00 | 0.43 | 0.00 | 1.07 | 0.90 | 1.01 | 0.52 |
| 2004 | 0.28 | 0.06 | 0.22 | 0 | 0.17 | 0.00 | 0.17 | 0.00 | 1.13 | 0.92 | 0.96 | 0.46 |
| 2005 | 0.08 | 0.09 | 0.09 | 0 | 0.06 | 0.00 | 0.08 | 0.00 | 0.57 | 0.97 | 0.34 | 0.55 |
| 2006 | 0.11 | 0.07 | 0.15 | 0 | 0.12 | 0.00 | 0.12 | 0.00 | 0.72 | 0.90 | 0.44 | 0.45 |
| 2007 | 0.19 | 0.06 | 0.21 | 0 | 0.18 | 0.00 | 0.19 | 0.00 | 1.14 | 1.10 | 0.79 | 0.41 |
| 2008 | 0.21 | 0.08 | 0.26 | 0 | 0.22 | 0.00 | 0.21 | 0.00 | 1.18 | 1.03 | 0.77 | 0.61 |
| 2009 | 0.12 | 0.07 | 0.16 | 0 | 0.13 | 0.00 | 0.12 | 0.00 | 0.82 | 0.99 | 0.52 | 0.63 |
| 2010 | 0.23 | 0.11 | 0.31 | 0 | 0.22 | 0.01 | 0.22 | 0.00 | 1.00 | 1.08 | 0.61 | 0.66 |
| 2011 | 0.17 | 0.10 | 0.22 | 0 | 0.17 | 0.00 | 0.16 | 0.00 | 1.00 | 0.92 | 0.54 | 0.57 |
| 2012 | 0.24 | 0.10 | 0.30 | 0 | 0.25 | 0.01 | 0.24 | 0.01 | 0.70 | 0.91 | 0.49 | 0.69 |
| 2013 | 0.20 | 0.08 | 0.28 | 0 | 0.20 | 0.00 | 0.20 | 0.00 | 0.70 | 0.86 | 0.45 | 0.54 |
| 2014 | 0.15 | 0.07 | 0.19 | 0 | 0.15 | 0.00 | 0.15 | 0.00 | 0.43 | 0.94 | 0.35 | 0.51 |
| 2015 | 0.18 | 0.07 | 0.21 | 0 | 0.17 | 0.00 | 0.17 | 0.00 | 0.61 | 0.87 | 0.48 | 0.48 |
| 2016 | 0.09 | 0.07 | 0.04 | 0 | 0.04 | 0.01 | 0.04 | 0.00 | 0.43 | 1.01 | 0.34 | 0.45 |
| 2017 | 0.21 | 0.09 | 0.04 | 0 | 0.06 | 0.00 | 0.04 | 0.00 | 0.68 | 0.91 | 0.57 | 0.55 |
| 2018 | 0.14 | 0.07 | 0.15 | 0 | 0.15 | 0.00 | 0.15 | 0.00 | 0.51 | 0.87 | 0.34 | 0.54 |
| 2019 | 0.34 | 0.06 | 0.28 | 0 | 0.25 | 0.00 | 0.19 | 0.00 | 1.00 | 0.89 | 0.78 | 0.56 |
| 2020 | 0.53 | 0.10 | 0.90 | 0 | 0.28 | 0.00 | 0.27 | 0.00 | 0.89 | 0.94 | 0.67 | 0.61 |
| 2021 | 0.38 | 0.07 | 0.45 | 0 | 0.34 | 0.00 | 0.13 | 0.00 | 0.88 | 0.82 | 0.76 | 0.53 |
| 2022 | 0.26 | 0.08 | 0.34 | 0 | 0.27 | 0.00 | 0.26 | 0.00 | 0.82 | 0.93 | 0.63 | 0.60 |


We selected four machine learning methods that prediction is closer to the training
target of sea surface salinity (with the mean RMSE less than 0.25), which are RF, KNN,
LGB, and CB. These four machine learning methods have better prediction results for
EN4 than for CTD. The errors generated during the prediction process mainly come
from the prediction of CTD salinity. The annual differences in predictive capabilities of
these four types of machine learning are very significant. The prediction results for RF
were the best in 2005 and 2016, and the worst in 2020, KNN had the best prediction
results for 2016 and 2017, and the worst prediction results for 2020. LGB had the best
forecast results for 2016 and 2017, and the worst forecast results for 2003. CB had the
best forecast results for 2016 and 2017, and the worst forecast results for 2003. In the
same year, some machine learning predictions are good while others are poor. For
example, in 2020, the predictions of RF and KNN were poor, but the predictions of
LGB and CB were good. This indicates that using multiple machine learning methods
can help improve the predictions of a certain method that performed poorly in a
particular year, eliminate biases in selecting machine learning methods for predictions,
and make the predictions more reliable.





RMSE is the spatial average result (Table 1), so only considering the numerical value
of RMSE will ignore the predictive ability of machine learning methods on different
regions in space. After training, we selected four machine learning methods with the
mean RMSE less than 0.25, which are RF, KNN, LGB, and CB. We take the example
of the prediction error of surface salinity in 2008 (predicted value minus training target
value) to analyze the salinity prediction ability of machine learning methods in different
regions. Machine learning models has significant spatial differences in predicting
salinity of CTD. Specifically, there are larger prediction errors in the Chukchi Sea,
Chukchi Sea Shelf, southern continental shelf slope of the Beaufort Gyre and center
Canada basin. The largest error occurred in the Chukchi Sea, which may be due to the
influence of Pacific water on the salinity of the upper layer of the Western Arctic Ocean.
The four machine learning methods for predicting surface salinity in EN4 are all very
good. KNN, LGB, and CB even have negligible prediction errors. RF shows a
significant spatial distribution in predicting surface salinity in EN4, with
overestimations in the southeast of the Canadian Basin and the western part of the East
Siberian Sea, with prediction errors less than 0.2psu. The predictions are
underestimated in the Chukchi Sea and the East Siberian Sea. The prediction errors of
different machine learning methods vary, so different weights need to be considered in
the data mergence process.

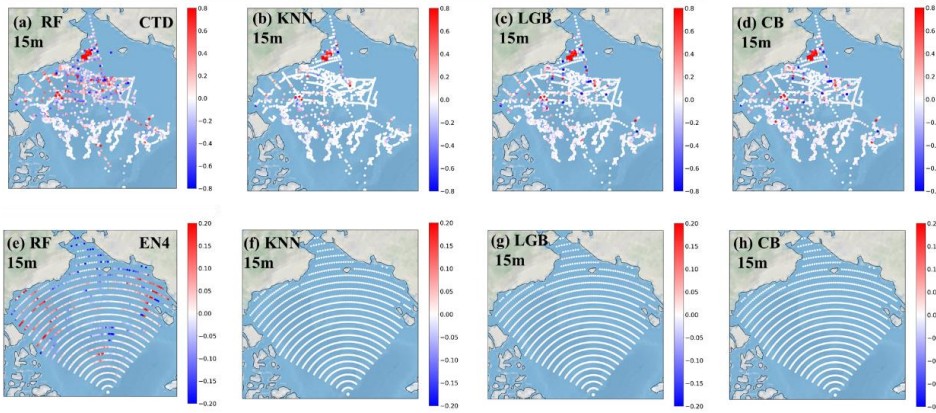


**Figure 5. Error between the predicted salinity and real salinity values for the**
**training pool in 2008.**
**2.4 Data merging and post-calibrating**
The third part is the data mergence part, where we linearly merging the training results
of the four better machine learning models. MAE is the average absolute difference



between the in situ data (true values) and the model output (predicted values). The sign
of these differences is ignored so that cancelations between positive and negative values
do not occur. RMSE and MAE have primarily been used to represent the uncertainties
in reconstructed datasets. In this article, we choose MAE as the criterion for assessing
uncertainty. We introduced weights and defined uncertainty, with uncertainty less than
2% as the indicator for selecting weights $a_{kj}$. The uncertainty (CT1) is calculated as
follows: $CT_{1kj} = \frac{1}{4}\sum_{k=1}^{4}\frac{|\hat{y}_{kj}-Z_j|}{Y_j}\times 100\%$ $(Eq.\ 2)$, Where k represents different
machine learning algorithms, and there are six types in total, which are RF, KNN, LGB,
CB; j=1 represents CTD data, j=2 represents EN4 data; $Z_j = \sum_{k=1}^{4}a_{kj}\hat{y}_{kj}$ $(Eq.3)$.
From this, we obtain the initial predicted products.
The salinity product is generated through the fourth post-calibrating, when there are
CTD measured data around the grid point, the salinity value of the point is formed by
merging the EN4 prediction results and the CTD prediction results according to weights;
otherwise, the salinity value of the point is taken as the EN4 prediction result. We
introduced weights and defined uncertainty, with uncertainty less than 3% as the
indicator for selecting weights $\beta_{kj}$. We need to check that salinity product $\hat{S} =$
$\sum_{j=1}^{2}\beta_j\hat{Z}_j$ $(Eq.4)$ by uncertainty judging. The uncertainty (CT2) is calculated as
follows: $CT_{2j} = \frac{1}{2}\sum_{j=1}^{2}\frac{|\hat{Z}_j-\hat{S}|}{\hat{S}}\times 100\%$ $(Eq.5)$, Where j=1 represents CTD data, j=2
represents EN4 data. From this, we obtain the final salinity product in the Western
Arctic Ocean.
The uncertainty of the data in this article (represented by rMAE) includes three parts:
one part is the uncertainty generated during the machine learning process, with an
uncertainty of 0.24% for the surface salinity prediction generated by CTD and 0.02%
for the surface salinity prediction generated by EN4; the other parts include
uncertainties in data merging (Fig. 6a, 6b) and post calibrating (Fig. 6c). There are two
sets of initial predicted products for data merging of machine learning methods, EN4
and CTD. The uncertainty generated shows that the uncertainty constrained by CTD
data is larger in the central part of the Canadian Basin and the Chukchi Sea Shelf and
its adjacent waters, reaching 1.63% in the central part of the Canadian Basin. The
uncertainty constrained by EN4 data is larger in the central part of the Canadian Basin
and the East Siberian Sea, reaching 0.44% in the East Siberian Sea. The uncertainty
generated during the post-calibrating process is highest in the Canadian basin, with a
maximum value of 2.54%.

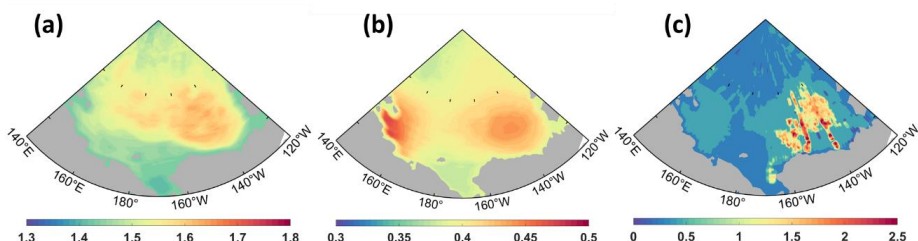


**Figure 6. Spacial pattern of sea surface salinity uncertainty (%) during the data**
**merging (a, CTD; b, EN4) and post-calibrating.**
**3. Result and Discussion**
We used the salinity product to calculate the freshwater content in the Beaufort Gyre
region (black box in Fig. 7a). In order to verify the superiority of the generated salinity
data in calculating the freshwater content, in addition to the freshwater content data
provided by BGEP for verification. On the other hand, the research of Hall et al. (2022)
showed that the salinity of ORAS5 and EN4 can be used to calculate the freshwater
content of the Arctic Ocean, and we also introduced the results of the freshwater content
calculation of ORAS5 (Fig .7b). The FWC was computed relative to salinity 34.8 psu
following Proshutinsky et al. (2009): $\text{FWC} = \int_{z34.8}^{zsurface} \left( \frac{34.8 - s(z)}{34.8} \right) dz \quad (Eq. 6)$
The absolute errors of the freshwater content calculated by the generated salinity
product, the salinity data of EN4 and ORAS5 and the freshwater content provided by
BGEP are 4.89%, 13.21% and 16.40%, respectively. Using the generated salinity
product to calculate the freshwater content in the Beaufort Gyre region area can
improve the accuracy. We compared the spatial distribution of freshwater content
calculated from salinity product with freshwater content provided by BGEP. There are
areas on the Mendeleev Ridge with large freshwater content, which may be formed by
fresh water advection from the East Siberian Sea or by freshwater advection from the
Beaufort Gyre.

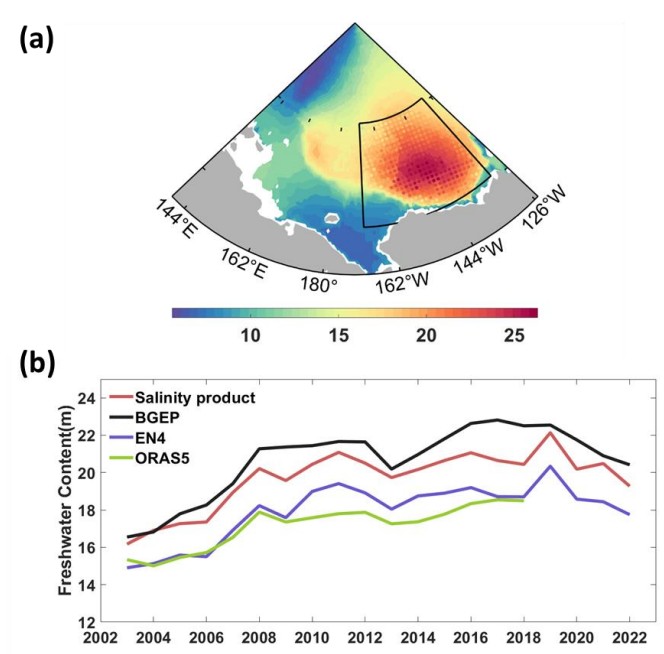


**Figure 7. Temporal and special variation of Freshwater Content (FWC, m). (a)Shadow is Mean FWC from 2003 to 2022 derived from salinity product, color dots represent FWC provided by BGEP. (b) Time series of FWC in Beaufort Gyre region, Beaufort Gyre region is the black box in (a).**


The depth of halocline base plays an important role in studying the Beaufort Gyre
dynamics (e.g. Manucharyan et al.,2016). The depth of the halocline base is determined
by taking the 33.9 psu isosalinity line (Lin et al.,2023; Nyugen et al.,2012). All salinity
data used were interpolated vertically to 2m to calculate the depth of the halocline base.
The salinity product, EN4, ORAS5 and WOA18 calculated the halocline base depth in
Beaufort Gyre region of 192m,191m,187m and 176m, respectively (Fig. 8d). Salinity
product allow more accurate calculation of depth of halocline depth. Compared with
the results of ORAS5, the depth of halocline calculated by salinity product increased
significantly in the 2000s. Compared with EN4 results, the deepening trend in the 2010s
is more significant, but smaller than that of ORAS5. We compared the spatial
distribution characteristics of the bottom halocline and WOA18 obtained from salinity
product. The depth of halocline base is the deepest in the Canadian Basin, but the
salinity product results are shallower and more easterly than WOA18. The depth of the
halocline base calculated by salinity product is obviously 21m shallower in the
southwest of the Canadian Basin and 23m deeper in the north of the East Siberian Sea

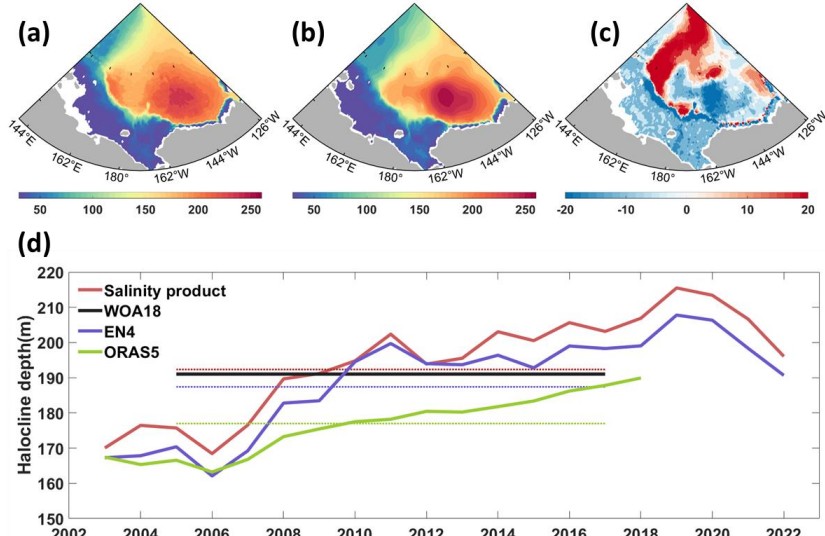


**Figure8. Temporal and special variation of Halocline depth (m). (a)Mean halocline depth from 2005 to 2017 derived from salinity product (b) Mean halocline base depth from 2005 to 2017 derived from salinity of WOA18. (c) Mean halocline depth difference between salinity product and WOA18 from 2005 to 2017. (d) Time series of halocline depth in Beaufort Gyre region.**

The results of salinity product indicate that the surface salinity is characterized by low salinity in the central Canadian Basin and the East Siberian Sea, which indicates the accumulation of fresh water there (Fig. 9). The continuous decrease in surface salinity before 2011 and the continuous increase in surface salinity after 2011 indicate that freshwater accumulated mainly at the surface before 2011 and decreased after 2011, which support the recent major freshening event from 2012 to 2016 in North Atlantic (Holliday et al.,2020). In the east-west direction, the surface low salt characteristics westward expanded from 2003 to 2013, and eastward from 2014 to 2022, which supports the conclusion that Beaufort Gyre expands westward (Regan et al.,2019; Armitage et al., 2017) and shrinks eastward (Lin et al.,2023). In the north-south direction, the surface low salt characteristics expanded northward in 2007, 2008, 2015 and 2016. The surface salinity of the East Siberian Sea decreased significantly in 2008 and has remained at reduced levels since then. According to the characteristics of surface ocean circulation (Armitage et al., 2017), surface freshwater in the East Siberian

Sea may enter the Beaufort Gyre or flow out of the Arctic Ocean along the transpolar
drift. The characteristics of sea surface salinity can be seen that the Pacific water flows
partly to the northern Chukchi Sea, partly to the Canadian Basin and partly to the CAA
along the Alaskan coastal current, the reduced sea surface salinity of the Alaskan coastal
current indicates that less Pacific water is being transported along this path, indicating
a weakening of the Alaskan coastal current, whether this is influenced by the enhanced
Beaufort Gyre.

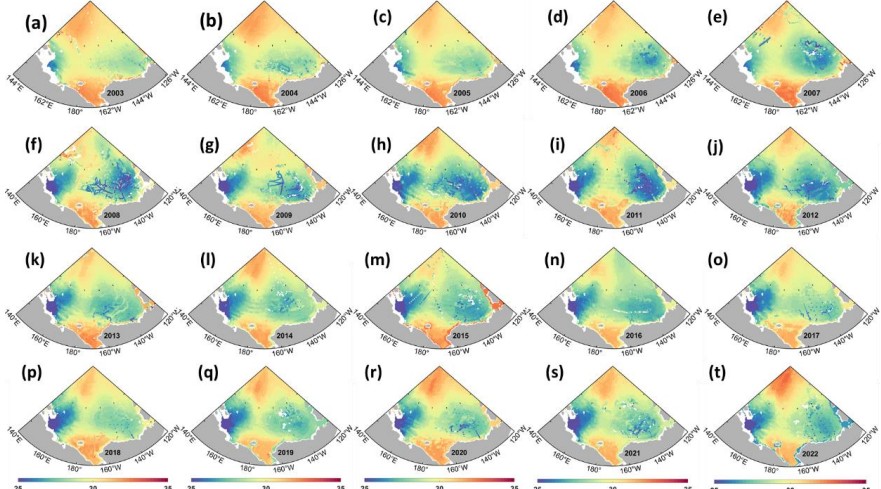


**Figure 9. Annual sea surface salinity fields in the Western Arctic ocean from 2003**
**to 2022. The color dots represent the measured CTD results, and the white dots**
**represent the measured sites that were deleted after quality control (see section**
**2.2).**
In order to observe the salinity distribution at the bottom of the halocline, which is about
200m deep in the western Arctic Ocean, we have analyzed the salinity distribution at
200m (Fig. 10). The results of salinity product indicate that salinity at 200m is
characterized by low salinity in the central Canadian Basin which indicates the
accumulation of fresh water in Canada Basin. Unlike the sea surface salinity, the salinity
at 200m has remained a slow downward trend after a rapid decline before 2008. This
suggests that fresh water in the Canadian Basin was relatively stable after a rapid
accumulation prior to 2008. Prior to 2008, freshwater in the western Arctic Ocean
pooled in large quantities at both the surface and the bottom of the halocline. After 2008,
the surface water decreased significantly while the bottom of the halocline water still
increased, indicating that the freshwater may be redistributed in the Arctic Ocean





through westward and northward expansion into the Marklov Basin (Bertosio et
al.,2022) or transported out of the Arctic Ocean (Zhang et al.,2021), or it may be pooled
deeper into the water column. From 2003 to 2013, the range of low salinity
characteristics of the halocline depth expanded, indicating that the area of freshwater
reservoir expanded and the area of Beaufort Gyre expanded. The salinity at 200m in
2022 increases significantly, indicating that there may be a freshwater migration
process in 2022.

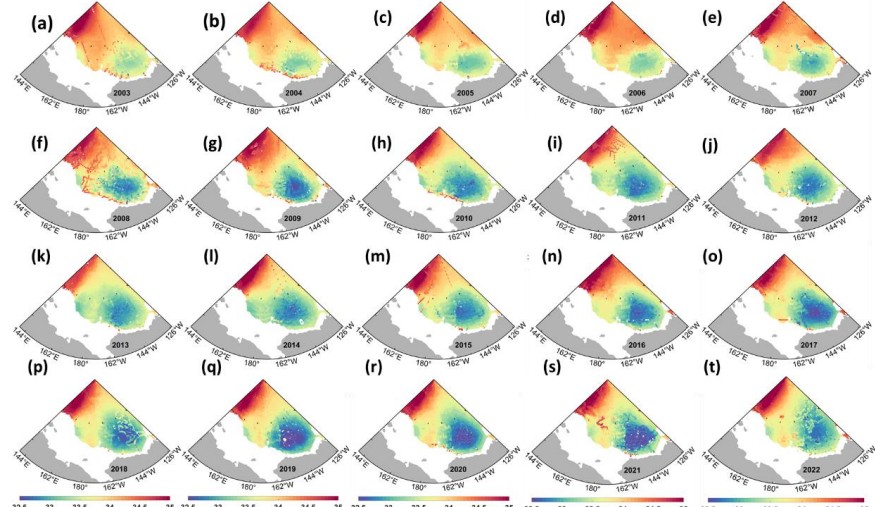

**Figure 10. Reconstructed annual salinity fields at 200m in the Western Arctic
ocean from 2003 to 2022.**

**4. Data availability**

The salinity product (0.5×0.25°, 2003-2022) is available at
https://zenodo.org/records/10990138 (Tao and Du, 2024).

**5. Summary**

Based on data mining-based machine learning method, we have provided a salinity
product for the Western Arctic Ocean with a resolution of 0.5°×0.25° for the period
spanning from 2003 to 2022. This was achieved by establishing correlations between
bathymetry, sea ice dynamics, atmospheric conditions, and seawater salinity. The input
variables employed in our machine learning model encompass ERA5 data (sea level
pressure), NSIDC information (sea ice concentration and motion), as well as ETOPO1

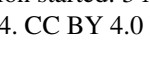



dataset (bathymetric details). After filtering, we employ four machine learning
algorithms (Random Forest, K Nearest Neighbor, LightGBM, CatBoost) to train
salinity data obtained from EN4 and CTD. Utilizing multiple machine learning methods
can mitigate the impact of inherent flaws in a specific method on the results. During
data integration, varying weight combinations of variables greatly affect uncertainty;
therefore, we implement an uncertainty threshold to constrain appropriate weights.
We conducted an analysis to determine the significance of five input variables in
predicting salinity, which serves as a reliable indicator for identifying the key factors
influencing salinity changes. However, it is crucial to acknowledge that there might be
potential interactions among different variables. The importance of various factors
varies when predicting salinity in both EN4 and CTD datasets. Interestingly, both
datasets consistently highlight sea level pressure as the primary influential factor for
surface salinity prediction, while sea ice concentration emerges as the main determinant
when forecasting salinity at a depth of approximately 200m (corresponding to the
halocline base). The impact of sea ice movement on the surface is more significant than
that on the bottom of the halocline. The meridional ice speed is advantageous for
salinity prediction using CTD data, while the zonal flow speed is advantageous for
salinity prediction using EN4 data. However, the contribution of water depth factors
varies. CTD data indicates that water depth has a dominant influence on salinity
prediction in deep layers, whereas EN4 data shows the opposite trend. Salinity is closely
associated with freshwater distribution. The transport and accumulation of surface
freshwater are regulated by the sea level pressure field, and the melting of sea ice exerts
a greater impact on salinity compared to its movement affecting freshwater.

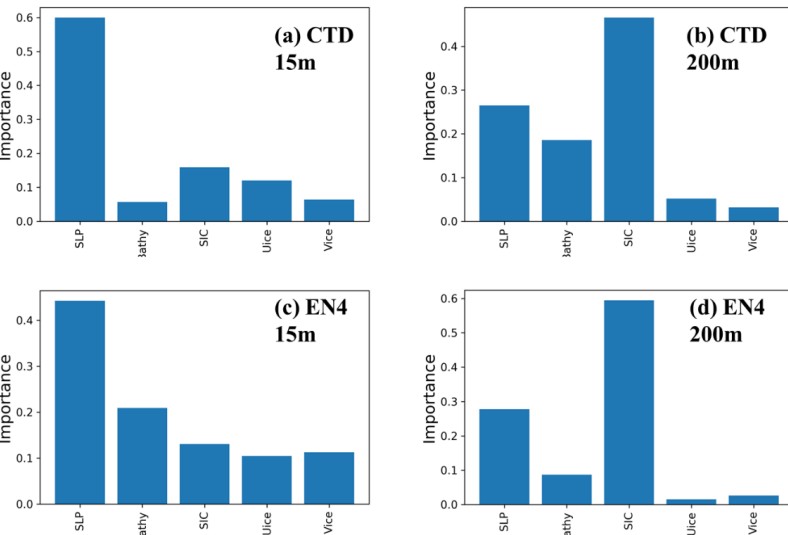



**Figure 11. Importance of different input variance.**

Accurate salinity product is crucial for understanding the dynamics of the Beaufort Gyre and the redistribution of freshwater in the Beaufort Gyre in the western Arctic Ocean. Hall et al. (2022) demonstrated that EN4 and ORAS5 salinity data can be utilized for Arctic Ocean studies. However, when compared to EN4 and ORAS5, salinity-derived freshwater content aligns more closely with BGEP estimates, suggesting superior accuracy in FWC calculations. Furthermore, considering the precision depth of halocline base, salinity products exhibit greater accuracy than EN4 and ORAS5. The findings from salinity product reveal a significant increase in freshwater content throughout the upper 200m layer of the Beaufort Gyre during the 2000s; however, surface freshwater decreased while subsurface fresh water continued to accumulate during the 2010s. It is likely that surface fresh water has been redistributed towards Marklov Basin (Bertosio et al., 2022), potentially accumulating in subsurface layers due to Ekman Pumping influences.

The salinity field of the Western Arctic Ocean is taken as an example to construct a novel data mining method for polar sea areas, utilizing multiple machine learning methods that integrate multiple data sources and incorporate physical processes. The application potential of this method extends beyond the salinity field and includes other related fields like hydrometeorology, sea ice thickness, polar biogeochemistry, among others. It effectively utilizes multi-machine learning results for data evaluation and integration.

**Author contributions.** LD provided scientific ideas, reviewed the paper and contributed to the revising of figures and words of this paper; ST collected the datasets, wrote the codes, analyzed the data, plotted the figures and wrote the paper. JL contributed to the revising of figures and words of this paper

**Competing interests.** The contact author has declared that none of the authors has any competing interests.

**Disclaimer.** Publisher's note: Copernicus Publications remains neutral with regard to jurisdictional claims made in the text, published maps, institutional affiliations, or any other geographical representation in this paper. While Copernicus Publications makes every effort to include appropriate place names, the final responsibility lies with the authors.

**Acknowledgements.** We collected a large number of public data, collected during




numerous expeditions and cruises over several decades. It is impossible to mention all
researchers who have contributed to these projects. We therefore thank all these people
for the gigantic effort, the huge amount of work and for making their data freely
available. This work was supported by the National Natural Science Foundation of
China (grant no. 42230405, 41976217, and 41576020) and the Global Change Research
Program of China (No. 2015CB953902).

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
