# Peer review of "Data mining-based machine learning methods for improving"

_Earth System Science Data, 2024_

## Author Comment (AC1)

'No signs of cross-validation against independent data, probable overfitting'

The salinity dataset by Tao and co-authors is interpolating salinity profiles in the data-scarce Western Arctic Ocean with the help of an interpolated salinity database and auxiliary atmospheric and sea ice observations. As many as six different machine learning algorithms are used to merge the various data. The resulting data product is used to calculated freshwater contents and compared to in situ and model-based estimates.

The major weakness of the paper is the lack of consideration for the validation against independent data, both in the choice of data sources and then in the machine learning methodology. The authors use different data sources (WOD, EN4, UDASH and the WOA18 climatology) without indication of the inter-dependences between these datasets. Whether the CTD data has been included in EN4 or not has implications on how I am interpreting the results: is the product an improved interpolation method than the objective analysis used in EN4 or are the two predictions of CTD and EN4 values two independent estimates of the same salinities? The same question applies for the BGEP mooring data, are they included or not in the EN4, UDASH and WOA18 aggregators? This question is an important prerequisite to understand why the estimates differ so much in the results section. As the paper stands, these results are just numbers that could look right for the wrong reason.

**[About Cross-validation]** First of all, thank you for dedicating your valuable time and effort to conduct a thorough review. We supplement the K-fold cross-validation during the machine learning training process in the training pool. We used R2 to verify the ability of machine learning training (below). We selected RF, KNN, LGB, CB based on cross validation results, which is same as the original manuscript.

Table. Results (R2) of cross-validation of training pool.

| | RF | | KNN | | LightGBM | | Catboost | | Multilinear Regression | | Neural Network | |
| --- | --- | --- | --- | --- | --- | --- | --- | --- | --- | --- | --- | --- |
| | CTD | EN4 | CTD | EN4 | CTD | EN4 | CTD | EN4 | CTD | EN4 | CTD | EN4 |
| 2003 | 0.61 | 0.48 | 0.44 | 0.39 | 0.57 | 0.39 | 0.54 | 0.49 | 0.31 | -0.55 | -0.21 | -0.21 |
| 2004 | 0.75 | 0.66 | 0.73 | 0.17 | 0.71 | 0.56 | 0.71 | 0.56 | -0.36 | -0.38 | -0.14 | -0.14 |
| 2005 | 0.60 | 0.20 | 0.49 | 0.29 | 0.14 | -0.29 | 0.42 | -0.03 | -0.50 | -0.39 | -0.24 | -0.24 |
| 2006 | 0.88 | 0.62 | 0.86 | 0.18 | 0.90 | 0.46 | 0.85 | 0.52 | 0.19 | -0.24 | -0.36 | -0.36 |
| 2007 | 0.80 | 0.78 | 0.80 | 0.06 | 0.82 | 0.56 | 0.77 | 0.66 | 0.16 | -0.40 | -0.44 | -0.44 |
| 2008 | 0.81 | -0.40 | 0.78 | -0.19 | 0.77 | -0.79 | 0.79 | 0.03 | 0.15 | -1.21 | -1.04 | -1.04 |
| 2009 | 0.86 | 0.41 | 0.78 | -0.38 | 0.60 | 0.02 | 0.85 | 0.16 | 0.57 | -0.41 | -0.34 | -0.34 |
| 2010 | 0.79 | 0.31 | 0.79 | -0.10 | 0.80 | -0.01 | 0.73 | 0.39 | 0.36 | -0.52 | -0.37 | -0.37 |
| 2011 | 0.95 | 0.27 | 0.92 | 0.27 | 0.90 | 0.16 | 0.84 | 0.27 | 0.52 | -0.21 | -0.67 | -0.67 |
| 2012 | 0.76 | 0.12 | 0.87 | 0.27 | 0.86 | -0.05 | 0.87 | 0.21 | 0.61 | 0.03 | -0.33 | -0.33 |
| 2013 | 0.63 | 0.25 | -0.28 | -0.09 | 0.51 | 0.15 | 0.55 | 0.41 | -0.13 | -0.27 | -0.27 | -0.27 |
| 2014 | -0.85 | 0.49 | -0.35 | 0.22 | -0.54 | 0.48 | -0.52 | 0.51 | -0.21 | -0.18 | -0.41 | -0.41 |
| 2015 | -0.73 | 0.45 | -0.95 | -0.03 | -0.81 | 0.43 | -1.04 | 0.50 | -0.33 | -0.39 | -0.49 | -0.49 |
| 2016 | -0.70 | 0.50 | -0.82 | -0.26 | -0.41 | 0.41 | -0.27 | 0.50 | -0.42 | -2.08 | -0.67 | -0.67 |
| 2017 | -0.98 | 0.55 | -1.58 | 0.21 | -1.51 | 0.42 | -1.90 | 0.65 | -1.93 | -0.63 | -0.15 | -0.15 |
| 2018 | -0.01 | 0.54 | -2.30 | 0.32 | -1.46 | 0.61 | 0.20 | 0.64 | -1.30 | -0.06 | -0.24 | -0.24 |
| 2019 | -0.65 | 0.70 | -2.22 | 0.13 | -10.32 | 0.68 | -0.36 | 0.66 | -4.84 | 0.04 | -0.36 | -0.36 |
| 2020 | 0.08 | 0.51 | -0.08 | 0.09 | -0.13 | 0.48 | -0.06 | 0.51 | -0.14 | 0.10 | -0.17 | -0.17 |
| 2021 | 0.14 | 0.36 | -0.14 | 0.34 | -0.23 | 0.30 | -0.08 | 0.39 | -0.23 | 0.08 | -0.20 | -0.20 |
| 2022 | 0.05 | 0.38 | -1.13 | 0.30 | -0.24 | 0.39 | -0.29 | 0.67 | -0.06 | -0.01 | -0.27 | -0.27 |
| Average | 0.24 | 0.41 | -0.12 | 0.11 | -0.40 | 0.27 | 0.18 | 0.44 | -0.38 | -0.38 | -0.37 | -0.37 |

The CTD data was utilized in this article, which includes the ITP data, LSSL data, WOD data and UDASH data. The spatial distribution of CTD data from these different sources exhibits some degree of overlap, yet notable disparities persist (refer to Figure 1). The salinity product of this paper is not an improved interpolation method than the objective analysis used in EN4. We utilized the scattered CTD data (see Figure1) alongside the gridded EN4 objective analysis results to generate two sets of predicted gridded data employing a post-filtered machine learning approach, subsequently merging these two sets into the ultimate salinity product.

The BGEP datasets utilized in this study primarily consist of LSSL CTD data for machine learning training and grid-based freshwater content data for validating the accuracy of salinity product calculations. The two sets of data are different. The LSSL CTD datasets (2003-2020) from BGEP, represented by purple dots, also show some degree of overlap; however, notable disparities persist when compared to UDASH (2003-2015) and WOD18 (2003-2020). Although EN4 and UDASH salinity datasets contains part BGEP LSSL CTD data, but they also contain other sources of data, is used to generate the salinity of the grid data of different methods.

[Figure]

Dots indicate the locations of observational site from different sources.

BGEP freshwater content is the Estimation of liquid freshwater content (FWC) of the Beaufort Gyre region (BGR) are computed following Proshutinsky et al. (2009) using CTD, XCTD, and UCTD profiles collected each year. The FWC is calculated using optimal interpolation on a 50-km square grid between 70˚N and 80˚N, and 130˚W - 170˚W, and where water depths exceed 300 m. The salinity product can be compared with EN4, ORAS5, BGEP gridded freshwater.

[Figure]

FWC in the BGR based on hydrographic measurements in 2003, The black dots indicate the locations of observational sites.(Figure from https://www2.whoi.edu/site/beaufortgyre/data/freshwater-content-gridded-data/)

The methodological concern is that all six machine learning techniques used will overfit the salinity data unless a part (say, one or two years) are set aside for validation. This is common practice in machine learning as all textbooks will show. From the text and the columns of zeroes in Table 1, the results are presented on the training data, which is therefore not guaranteeing any skills in extrapolation. Along the same lines, the uncertainty CT1 is computed from the residuals of the training datasets, which in the likely case of overfitting are much lower than the actual errors (and worse, unrelated to them). So if there were one additional CTD cast that was not included in the input dataset, I see no guarantee that the prediction in that point would be as skillful as indicated in the paper. I would prefer one properly validated algorithm to six different ways of overfitting.

**[About probable overfitting]** First of all, we thank the reviewer for your professional questions. Overfitting is indeed an emerging problem in machine learning. The problem of overfitting has been considered in the process of machine learning in this paper. The datasets used for prediction from each year were randomized, as depicted in Figure 4 of the text. Subsequently, 90% of the data was selected for training purposes, constituting the training pool, while the remaining 10% was allocated for testing purposes, forming the testing pool. After we control the overfitting, we firstly used threshold (0.25) of training(90%) pool to evaluate the prediction skill of different machine-learning methods, we selected four machine learning methods that prediction is closer to the training target of sea surface salinity (with the mean RMSE less than 0.25), which are RF, KNN, LGB, and CB. We added the verification results of testing pool (the table below), to get the same results. Therefore, the skills in extrapolation was guaranteed.

The predicted values of these different machine learning methods were the RMSE of the original and predicted values of the test pool (10%) controlled at a threshold of 0.25, and then

extrapolated to the predicted values of the production grid after confirming the reliability of the machine learning method. (line274-278). CT1 is our respectively based on EN4 data to a variety of machine learning and CTD data screening for the error of the predicted results of the first merging (figure 6 a,b). The predicted values of these different machine learning methods were the RMSE of the original and predicted values of the test pool (10%) controlled at a threshold of 0.25, and then extrapolated to the predicted values of the production grid after confirming the reliability of the machine learning method. The constraint of CT1 comprises two components, namely the merging uncertainty associated with multiple machine learning predictions for CTD data and the merging uncertainty related to multiple machine learning predictions for EN4 data. However, typically, the predictive value merging process of CTD data tends to generate a higher level of uncertainty. In practical operations, our focus lies in controlling the predictive value merging uncertainty of CTD to be lower than that of CT1.

| | RF | | KNN | | LightGBM | | Catboost | | Multilinear Regression | | Neural Network | |
|---|---|---|---|---|---|---|---|---|---|---|---|---|
| | CTD | EN4 | CTD | EN4 | CTD | EN4 | CTD | EN4 | CTD | | CTD | EN4 |
| 2003 | 0.42 | 0.05 | 0.34 | 0.00 | 0.34 | 0.00 | 0.32 | 0.00 | 1.07 | 1.00 | 1.00 | 2.01 |
| 2004 | 0.21 | 0.05 | 0.14 | 0.00 | 0.16 | 0.01 | 0.16 | 0.00 | 0.92 | 1.03 | 0.73 | 2.59 |
| 2005 | 0.14 | 0.11 | 0.12 | 0.00 | 0.13 | 0.01 | 0.13 | 0.00 | 0.65 | 1.07 | 0.40 | 2.03 |
| 2006 | 0.10 | 0.07 | 0.11 | 0.00 | 0.12 | 0.00 | 0.12 | 0.00 | 0.69 | 1.00 | 0.45 | 3.11 |
| 2007 | 0.21 | 0.05 | 0.14 | 0.00 | 0.20 | 0.01 | 0.20 | 0.00 | 1.11 | 1.21 | 0.91 | 3.54 |
| 2008 | 0.21 | 0.05 | 0.23 | 0.00 | 0.23 | 0.00 | 0.23 | 0.00 | 1.20 | 1.12 | 0.81 | 1.66 |
| 2009 | 0.11 | 0.05 | 0.14 | 0.00 | 0.11 | 0.00 | 0.12 | 0.00 | 0.81 | 1.01 | 0.51 | 2.03 |
| 2010 | 0.21 | 0.11 | 0.23 | 0.00 | 0.22 | 0.01 | 0.22 | 0.00 | 0.94 | 1.20 | 0.59 | 2.94 |
| 2011 | 0.18 | 0.07 | 0.18 | 0.00 | 0.19 | 0.00 | 0.17 | 0.00 | 0.99 | 1.03 | 0.66 | 1.74 |
| 2012 | 0.25 | 0.11 | 0.26 | 0.00 | 0.24 | 0.00 | 0.25 | 0.00 | 0.70 | 1.02 | 0.61 | 2.92 |
| 2013 | 0.18 | 0.08 | 0.17 | 0.00 | 0.18 | 0.00 | 0.18 | 0.00 | 0.72 | 0.99 | 0.44 | 2.49 |
| 2014 | 0.16 | 0.06 | 0.16 | 0.00 | 0.15 | 0.01 | 0.15 | 0.00 | 0.43 | 1.08 | 0.36 | 2.51 |
| 2015 | 0.23 | 0.07 | 0.17 | 0.00 | 0.27 | 0.00 | 0.21 | 0.00 | 0.71 | 0.97 | 0.66 | 1.96 |
| 2016 | 0.08 | 0.05 | 0.02 | 0.00 | 0.03 | 0.00 | 0.02 | 0.00 | 0.39 | 1.09 | 0.46 | 2.12 |
| 2017 | 0.11 | 0.05 | 0.03 | 0.00 | 0.04 | 0.00 | 0.02 | 0.00 | 0.53 | 1.00 | 0.46 | 2.87 |
| 2018 | 0.12 | 0.06 | 0.13 | 0.00 | 0.15 | 0.00 | 0.13 | 0.00 | 0.58 | 0.99 | 0.33 | 1.76 |
| 2019 | 0.33 | 0.05 | 0.25 | 0.00 | 0.50 | 0.00 | 0.22 | 0.00 | 0.82 | 0.96 | 0.77 | 1.92 |
| 2020 | 1.19 | 0.05 | 0.83 | 0.00 | 1.19 | 0.00 | 1.19 | 0.00 | 1.67 | 1.07 | 1.66 | 3.81 |
| 2021 | 0.69 | 0.07 | 0.69 | 0.00 | 0.69 | 0.00 | 0.69 | 0.00 | 1.54 | 0.88 | 1.80 | 2.57 |
| 2022 | 0.20 | 0.06 | 0.21 | 0.00 | 0.20 | 0.01 | 0.21 | 0.00 | 0.78 | 1.06 | 0.60 | 2.90 |
| Average | 0.27 | 0.07 | 0.23 | 0.00 | 0.27 | 0.00 | 0.25 | 0.00 | 0.86 | 1.04 | 0.71 | 2.47 |
| mean RMSE | 0.17 | | 0.11 | | 0.14 | | 0.12 | | 0.95 | | 1.59 | |

Overall the text is missing a clear explanation of the algorithm used and I had to squint at Figure 2 to imagine the methodology. The paper contains a lot of information, which relevance for the dataset is not mentioned. It overall makes a very tedious read and I realize that the correct procedures may have been applied without me finding it mentioned in the text.

In view of the above weaknesses, I believe the submitted manuscript cannot easily me modified into a publishable version. All the results should be presented on validation data rather than training data, and all the methods and data sections should be completely rewritten to specify the intended use of the data.

The results section should be rewritten to reflect on the reasons for the differences between EN4 and BGEP data and why the author's approach is the correct answer to the problem.

The expression of our idea in Figure 2 may not be ideal. Thank you for reminding us. We have redrawn Figure 2. The methods and data sections have been rewritten.

[Figure]

Figure 2 Procedure for improving the salinity field in the Western Arctic Ocean through a data mining-based machine learning method.

The results presented by training data are show in Figure5 and Table1, We added the verification results of testing pool (validation data) in the above table which indicate the similar result of Table1 in the manuscript. Thanks to your reminder, Table 1, which was based on training data, has now been replaced with results based on the test pool (validation data). At the same time, Figure 5 is the result of all the data (training data + testing(validation) data), we here supplement the statistical results based on the testing data in the table below. In

addition, all other results in the original manuscript are based on validation data and extrapolations after validation.

Table. MAE between the predicted salinity and target salinity of four selected machine learning methods in the testing pool

| | RF | | KNN | | LightGBM | | Catboost | |
|---|---|---|---|---|---|---|---|---|
| | CTD | EN4 | CTD | EN4 | CTD | EN4 | CTD | EN4 |
| 2003 | 0.22 | 0.03 | 0.12 | 0.00 | 0.14 | 0.00 | 0.13 | 0.00 |
| 2004 | 0.12 | 0.03 | 0.07 | 0.00 | 0.07 | 0.00 | 0.07 | 0.00 |
| 2005 | 0.05 | 0.04 | 0.03 | 0.00 | 0.04 | 0.00 | 0.04 | 0.00 |
| 2006 | 0.05 | 0.03 | 0.05 | 0.00 | 0.06 | 0.00 | 0.06 | 0.00 |
| 2007 | 0.07 | 0.03 | 0.06 | 0.00 | 0.06 | 0.00 | 0.06 | 0.00 |
| 2008 | 0.09 | 0.03 | 0.07 | 0.00 | 0.09 | 0.00 | 0.10 | 0.00 |
| 2009 | 0.05 | 0.03 | 0.05 | 0.00 | 0.06 | 0.00 | 0.05 | 0.00 |
| 2010 | 0.09 | 0.05 | 0.08 | 0.00 | 0.09 | 0.00 | 0.09 | 0.00 |
| 2011 | 0.09 | 0.04 | 0.07 | 0.00 | 0.09 | 0.00 | 0.08 | 0.00 |
| 2012 | 0.10 | 0.05 | 0.09 | 0.00 | 0.10 | 0.00 | 0.10 | 0.00 |
| 2013 | 0.07 | 0.03 | 0.06 | 0.00 | 0.06 | 0.00 | 0.06 | 0.00 |
| 2014 | 0.05 | 0.03 | 0.04 | 0.00 | 0.04 | 0.00 | 0.04 | 0.00 |
| 2015 | 0.09 | 0.03 | 0.04 | 0.00 | 0.08 | 0.00 | 0.06 | 0.00 |
| 2016 | 0.04 | 0.03 | 0.00 | 0.00 | 0.01 | 0.00 | 0.01 | 0.00 |
| 2017 | 0.06 | 0.03 | 0.01 | 0.00 | 0.02 | 0.00 | 0.01 | 0.00 |
| 2018 | 0.04 | 0.02 | 0.03 | 0.00 | 0.04 | 0.00 | 0.03 | 0.00 |
| 2019 | 0.11 | 0.03 | 0.07 | 0.00 | 0.11 | 0.00 | 0.07 | 0.00 |
| 2020 | 0.20 | 0.03 | 0.14 | 0.00 | 0.18 | 0.00 | 0.17 | 0.00 |
| 2021 | 0.11 | 0.04 | 0.06 | 0.00 | 0.08 | 0.00 | 0.08 | 0.00 |
| 2022 | 0.07 | 0.02 | 0.06 | 0.00 | 0.07 | 0.00 | 0.07 | 0.00 |
| Average | 0.09 | 0.03 | 0.06 | 0.00 | 0.08 | 0.00 | 0.07 | 0.00 |

What you mentioned that "the reasons for the differences between EN4 and BGEP data and why our approach is the correct answer to the problem" is not our main concern, but we have tried to supplement the discussion in this regard.

The high freshwater content provided by BGEP may be due to the fact that BGEP mainly uses summer (July-October) salinity data to calculate the freshwater content. Previous studies often used the freshwater content of BGEP to characterize the real freshwater content in the Beaufort Gyre Region (Proshutinsky et al.,2018; Zhang et al.,2021; Lin et al, 2023). Therefore, our salinity products only need to be closer to the freshwater content of BGEP than other data, which means that our salinity data is more accurate.

We used machine learning to reconstruct salinity based on the EN4. The results of the proposed comparison between optimal interpolation and machine learning methods are appended below. Compared with traditional methods, the machine learning method has the advantage of more accurate reconstruction of salinity. The MAE of EN4 salinity reconstruction by machine learning method (0.04psu) is significantly smaller than that by traditional method (0.09psu).

[Figure]

The mean salinity at 15m in the BG region based on EN4 (defined as BG box, Proshutinsky et al.,2009:170 ̊W-130 ̊W,70.5 ̊N-80.5 ̊N)

We incorporated optimal interpolation for reconstructing the CTD salinity, achieving an MAE of 0.73 psu. This value is slightly larger than the absolute error of 0.52 psu obtained from the machine learning.

[Figure]

The mean salinity at 15m in the BG region based on CTD (defined as BG box, Proshutinsky et al.,2009:170 ̊W-130 ̊W,70.5 ̊N-80.5 ̊N)

---

## Author Comment (AC2)

**审稿意见2**

First of all, thank you for dedicating your valuable time and effort to conduct a thorough review.

This paper employed multiple machine learning methods to reconstruct the salinity in the West Arctic Ocean based on easily obtained atmosphere reanalysis data and satellite-based sea ice concentration and motion. This topic is interesting and crucial, and this method can expand the spatial-temporal coverage and improve the accuracy of estimation than existing productions as the author claimed.

However, due to so much confusion about the method and verification, I have to treat this article with caution, and I believe that it now is far away from an adequate article, especially for publishing on ESSD. Here are some major comments.

- 1. For the method, there are 2 major questions.
  - 1. I cannot understand why the author using EN4 reanalysis data as a target to train his/her model. The EN4 including some uncertainties and errors cannot provide the true word's salinity. Moreover, EN4 can cover the whole area and time span this work focuses on. I think the author can use this dataset directly or simply interpolate it.

Firstly, EN4 data has been demonstrated that it can effectively analyze the salinity of the Arctic Ocean (*Hall et al.,2022*). The machine learning method is employed to train the EN4 data, with extrapolation serving three purposes on the salinity product grid: firstly, this article utilizes a machine learning approach to generate CTD data as well as EN4 data extrapolation results to generate salinity products by merging; secondly, it aims to demonstrate the selection process of input variables that can be utilized in machine learning and training; thirdly, the availability of CTD data in the east Siberian sea area is limited, therefore, the extrapolation results of salinity from EN4 data play a crucial role in generating the product.

2. I think the author wants to highlight the machine learnings, but I don't know which role they played in the improvement of the accuracy of salinity reconstruction. Cooperation with traditional methods (e.g., optimal interpolation) is lacking in this work. Moreover, the author compared their production with EN4, which is used as a target set to train the method, and the author said their results are better, which is contrary to general knowledge. Maybe the decrease in errors comes from the application of machine learning, but the more possible reason is just the merging of CTD data.

The salinity field of the Western Arctic Ocean is taken as an example to construct a

novel data mining method for polar sea areas in our paper, utilizing multiple machine learning methods that integrate multiple data sources and incorporate physical processes. We mainly used machine learning methods to train. We compared our production with EN4, and proved our results are better. Our training objective encompasses not only EN4 but also CTD, and the extrapolation results are merged to generate the salinity product. Therefore, salinity products in some aspects outperform EN4 without causing conflicts. As you said, the decrease in errors comes from the application of machine learning, and possible the merging of CTD data. The main focus of our research is to address the new method of data reconstruction in the polar regions.

2. For the verification, I feel very surprised about the 0 of RMSE for the results from the KNN method. It means that this method can perfectly reproduce your target salinity. The only possibility I can think of is that the author uses a train set the calculate RMSE instead of a verification or test set. This makes it completely impossible to evaluate the salinity reconstructed by the machine learning in this work.

First of all, we thank the reviewer for your professional questions. The datasets used for prediction from each year were randomized, as depicted in Figure 4 of the text. Subsequently, 90% of the data was selected for training purposes, constituting the training pool, while the remaining 10% was allocated for testing purposes, forming the testing pool. Indeed, we used a train set the calculate RMSE. So we added the verification results of testing pool (the table below), to get the same results. RMSE for the results from the KNN method is still about 0. Therefore, the skills in extrapolation was guaranteed.

|           | RF   |      | KNN  |      | LightGBM |      | Catboost |      | Multilinear
Regression |      | Neural
Network |      |  |
|-----------|------|------|------|------|----------|------|----------|------|---------------------------|------|-------------------|------|--|
|           | CTD  | EN4  | CTD  | EN4  | CTD      | EN4  | CTD      | EN4  | CTD                       |      | CTD               | EN4  |  |
| 2003      | 0.42 | 0.05 | 0.34 | 0.00 | 0.34     | 0.00 | 0.32     | 0.00 | 1.07                      | 1.00 | 1.00              | 2.01 |  |
| 2004      | 0.21 | 0.05 | 0.14 | 0.00 | 0.16     | 0.01 | 0.16     | 0.00 | 0.92                      | 1.03 | 0.73              | 2.59 |  |
| 2005      | 0.14 | 0.11 | 0.12 | 0.00 | 0.13     | 0.01 | 0.13     | 0.00 | 0.65                      | 1.07 | 0.40              | 2.03 |  |
| 2006      | 0.10 | 0.07 | 0.11 | 0.00 | 0.12     | 0.00 | 0.12     | 0.00 | 0.69                      | 1.00 | 0.45              | 3.11 |  |
| 2007      | 0.21 | 0.05 | 0.14 | 0.00 | 0.20     | 0.01 | 0.20     | 0.00 | 1.11                      | 1.21 | 0.91              | 3.54 |  |
| 2008      | 0.21 | 0.05 | 0.23 | 0.00 | 0.23     | 0.00 | 0.23     | 0.00 | 1.20                      | 1.12 | 0.81              | 1.66 |  |
| 2009      | 0.11 | 0.05 | 0.14 | 0.00 | 0.11     | 0.00 | 0.12     | 0.00 | 0.81                      | 1.01 | 0.51              | 2.03 |  |
| 2010      | 0.21 | 0.11 | 0.23 | 0.00 | 0.22     | 0.01 | 0.22     | 0.00 | 0.94                      | 1.20 | 0.59              | 2.94 |  |
| 2011      | 0.18 | 0.07 | 0.18 | 0.00 | 0.19     | 0.00 | 0.17     | 0.00 | 0.99                      | 1.03 | 0.66              | 1.74 |  |
| 2012      | 0.25 | 0.11 | 0.26 | 0.00 | 0.24     | 0.00 | 0.25     | 0.00 | 0.70                      | 1.02 | 0.61              | 2.92 |  |
| 2013      | 0.18 | 0.08 | 0.17 | 0.00 | 0.18     | 0.00 | 0.18     | 0.00 | 0.72                      | 0.99 | 0.44              | 2.49 |  |
| 2014      | 0.16 | 0.06 | 0.16 | 0.00 | 0.15     | 0.01 | 0.15     | 0.00 | 0.43                      | 1.08 | 0.36              | 2.51 |  |
| 2015      | 0.23 | 0.07 | 0.17 | 0.00 | 0.27     | 0.00 | 0.21     | 0.00 | 0.71                      | 0.97 | 0.66              | 1.96 |  |
| 2016      | 0.08 | 0.05 | 0.02 | 0.00 | 0.03     | 0.00 | 0.02     | 0.00 | 0.39                      | 1.09 | 0.46              | 2.12 |  |
| 2017      | 0.11 | 0.05 | 0.03 | 0.00 | 0.04     | 0.00 | 0.02     | 0.00 | 0.53                      | 1.00 | 0.46              | 2.87 |  |
| 2018      | 0.12 | 0.06 | 0.13 | 0.00 | 0.15     | 0.00 | 0.13     | 0.00 | 0.58                      | 0.99 | 0.33              | 1.76 |  |
| 2019      | 0.33 | 0.05 | 0.25 | 0.00 | 0.50     | 0.00 | 0.22     | 0.00 | 0.82                      | 0.96 | 0.77              | 1.92 |  |
| 2020      | 1.19 | 0.05 | 0.83 | 0.00 | 1.19     | 0.00 | 1.19     | 0.00 | 1.67                      | 1.07 | 1.66              | 3.81 |  |
| 2021      | 0.69 | 0.07 | 0.69 | 0.00 | 0.69     | 0.00 | 0.69     | 0.00 | 1.54                      | 0.88 | 1.80              | 2.57 |  |
| 2022      | 0.20 | 0.06 | 0.21 | 0.00 | 0.20     | 0.01 | 0.21     | 0.00 | 0.78                      | 1.06 | 0.60              | 2.90 |  |
| Average   | 0.27 | 0.07 | 0.23 | 0.00 | 0.27     | 0.00 | 0.25     | 0.00 | 0.86                      | 1.04 | 0.71              | 2.47 |  |
| mean RMSE | 0.   | 0.17 |      | 0.11 |          | 0.14 |          | 0.12 |                           | 0.95 |                   | 1.59 |  |

There are also many minor comments. What needs the author to pay more attention to is that many places in the MS are against the conventions of academic writing, which makes it hard to read.

Line 17: write the full name of "CTD" due to its first use in the paper. Similar issues exist in many other places, please check.

The expression of gratitude is extended to you for the reminder. the full name of "CTD" is conductivity, temperature, and depth.

The full text was carefully reviewed once again, and the following revise were taken. ITP (line 60, Ice-Tethered Profiler); CCGS (line139, Canadian Coast Guard Shipboard); NSIDC (line164, National Snow and Ice Data Center); MAE (line184, mean absolute error); FWC (line 293, Freshwater Content); WOA18 (World Ocean Atlas 2018); SLP (Sea Level Pressure).

Line 63: double full stops.

The redundant full stop has been removed, and a comprehensive inspection has been conducted.

Line 86: what's your dataset's temporal resolution?

Considering the spatial distribution of CTD data, the temporal resolution of salinity product is annual, disregarding seasonal fluctuations and focusing on interannual as well as lower frequency variations.

Line 96: I think it is better to put the section about why you focus on the Western Arctic Ocean in the Introduction than here.

We have put the reason we focus on the Western Arctic Ocean (Section 2.1 Study area original manuscript) in the introduction

Line 191: Fig. 2 is confused. It should be replotted. a), I cannot find the "Classify" and "statistic analysis" (case matters) in the text. b), Similarly, the "Physical process" and "Nearest Neighbors" (case matters) also cannot be found and they look like input variables very much. c), only 4 variables are used in the data-selecting step, which is far less than the data introduced in 2.2. And in the text, you don't seem to be doing anything with these 4 variables, while the CTD data was cleaned. This figure gives readers the opposite impression. You should make it clear to readers which are the variables used to train or build the dataset, and where are the algorithm. d), where is the WOA18 used when you create the dataset? Mark the figure or delete it.

The expression of our idea in Figure 2 may not be ideal. Thank you for reminding us. We have redrawn Figure 2. We mainly enriched the data selecting part and the machine learning part, and made some modifications to the data merging and post-calibrating.

Figure 2 Procedure for improving the salinity field in the Western Arctic Ocean through a data mining-based machine learning method.

(a) "classify" and "statistic analysis" are used to choose the final input variables. The process of "classify" and "statistic analysis" involves identifying the influential factors that impact the salinity of the arctic ocean, as discussed in previous literature. In terms of thermodynamics, the melting and freezing processes of sea ice have a significant effect on salinity. Additionally, from a dynamic perspective, both ice-ocean stress and air-ocean stress contribute to salinity redistribution within the ocean. Therefore, we propose incorporating variables such as sea level pressure(SLP), sea ice concentration(SIC), and sea ice drift speed(Uice,Vice).

(b) The "Physical process" and "Nearest Neighbors" are that "The salinity product is generated through the post-calibrating, when there are CTD measured data around the grid point, the salinity value of the point is formed by merging the EN4 prediction results and the CTD prediction results according to weights; otherwise, the salinity

value of the point is taken as the EN4 prediction result (line 258-261,original manuscript)". The post-calibrating included the concepts of "Physical process" and "Nearest Neighbors".

(c) The data introduced in 2.2 include CTD salinity data (from ITP, LSSL, WOD18, UDASH), EN4 salinity data, SLP (from ERA5), SIC (from NSIDC), Uice (from NSIDC), Vice (from NSIDC). The input variables include SLP, SIC, Uice, Vice. The output variables include Salinity (EN4 and CTD). So the data introduced in the 2.2 used to train(90%) and test(10%). Indeed, we don't do anything with these 4 variables, while the CTD data was cleaned. Because these 4 input variables provided by different institution are after Quality control. We think there are reliable. The time scale of the variables needs to be adjusted to align with that of CTD factors in machine learning. There is also ORAS5 data for the validation of salinity products as well as ORAS5 data.

Line 122 In this section, I think two things were done: 1, introduce the data used in this work; 2, data clean (selecting). It would be better to divide them into two paragraphs. A lot of data are introduced here, but it is confusing which one is used to train, which one is used to create the dataset, and which one is used to evaluate. Reorganize them according to their purpose.

Yes, the problem of overfitting has been considered in the process of machine learning in this paper. The datasets used for prediction from each year were randomized Subsequently, 90% of the data was selected for training purposes, constituting the training pool, while the remaining 10% was allocated for testing purposes, forming the testing pool. Your question is very meaningful, and we have added a summary at the end of 2.2

Line 127 "the data with flags 0 and 1 based on the quality control provided by the data itself": what does it mean?

WOD18 provides quality-controlled data, all data in the WOD are associated with as much metadata as possible, and every ocean data value has a quality control flag associated with it. Flag 0 means accepted value, Flag 1 means range outlier (outside of broad range check).

```
0: accepted value
1: range outlier (outside of broad range check)
2: failed inversion check
3: failed gradient check
4: observed level bullseye flag and zero gradient check
5: combined gradient and inversion checks
6: failed range and inversion checks
7: failed range and gradient checks
8: failed range and questionable data checks
9: failed range and combined gradient and inversion checks
```

Line 141: Why the data in 2004 are ignored?

The CTD data of LSSL collected during the 2004 expedition was not utilized. The potential temperature (shadow) and density values of the CTD data in 2004 are evidently anomalous (refer to the figure below), suggesting a potential issue with the data storage process, thus rendering them unsuitable for use.